# Experimental Suppression of Red Imported Fire Ants (*Solenopsis invicta*) Has Little Impact on the Survival of Eggs to Third Instar of Spring-Generation Monarch Butterflies (*Danaus plexippus*) Due to Buffering Effects of Host-Plant Arthropods

**Kalynn L. Hudman [1], Misty Stevenson [2], Kelsey Contreras [3], Alyx Scott [4] and Jeffrey G. Kopachena [1],\***

[1] Department of Biological and Environmental Sciences, Texas A&M University—Commerce, Commerce, TX 75428, USA
[2] Lakewood Country Club, Dallas, TX 75214, USA
[3] Cushman & Wakefield, 7701 Legacy Dr., Plano, TX 75024, USA
[4] Houston Arboretum & Nature Center, 4501 Woodway Dr., Houston, TX 77024, USA
**\*** Correspondence: jeff.kopachena@tamuc.edu

**Abstract:** The eastern migratory population of the monarch butterfly (*Danaus plexippus*) has shown evidence of declines in recent years. During early spring, when the population is at its smallest, red imported fire ants (RIFA) (*Solenopsis invicta*) have been implicated as having devastating effects on monarch egg and larval survival, but there are no conclusive experimental data to support this contention. The purpose of this study was to determine the effect of RIFA on the survival of spring monarch eggs to third instar larvae. Three treatments were analyzed: control plots, RIFA-suppressed plots, and RIFA-enhanced plots. Other host-plant arthropods were also documented. In control plots, monarch survival was unrelated to RIFA abundance on or around the plants. For both years combined, RIFA suppression had little impact on monarch survival. In one of the two years, higher survival occurred in the suppressed treatment, but confidence in this difference was low. In control plots, monarch survival increased with increasing numbers of other arthropods (not including RIFA) on the host plant. Predator pressure did not vary relative to arthropod abundance, and RIFA only occupied plants in large numbers when large numbers of other arthropods were also present. The presence of RIFA did not affect predator pressure. RIFA artificially drawn onto host plants created artificially high predator pressure, and monarch survival was low. Long-term use of bait to control RIFA may not be cost-effective provided surrounding biodiversity is high. Efforts to promote spring monarchs should focus on promoting biodiversity in addition to planting milkweed.

**Keywords:** butterfly; larval survival; milkweed community; predator pressure; indirect effects; prey population; ant control

## 1. Introduction

The monarch butterfly (*Danaus plexippus*) is an iconic butterfly whose populations in North America have shown evidence of dramatic population declines. These declines have resulted in a petition for listing as an endangered species by the U.S. Fish and Wildlife Service [1,2]. More recently, the eastern migratory population of monarch butterflies in North America was classified as endangered by the International Union for Conservation of Nature (IUCN) [3]. Based on winter roost data, the IUCN noted that the eastern migratory population had declined 84% between 1996 and 2014.

The eastern migratory population of monarch butterflies breeds through most of North America east of the Rocky Mountains, with a distribution that covers over 12 million km$^2$ [4]. The monarch butterfly is well-recognized for its dramatic migratory behavior; in the fall, millions of monarchs migrate as far as 4000 km to spend the winter in the mountains of central Mexico [5]. Over 85% of the eastern migratory population of monarchs

spend the winter in these winter roosts. In the spring, the overwintered adults migrate north, colonizing much of North America in a series of successive generations [4]. Owing to overwinter mortality, the adults leaving the winter roost sites represent the smallest population in the annual cycle [6]. Because of this, the first spring generation of monarchs is critically important in initiating the repopulation of much of North America east of the Rocky Mountains in subsequent generations [4]. Overwintered adults do not migrate far; almost all of the eggs laid by these individuals to produce the spring generation occur in the states of Texas and Oklahoma [4].

The monarch butterfly is a milkweed specialist; its larvae develop only on species of milkweed in the genus *Asclepias* along with a few closely related plants [7]. In eastern North America, the monarch butterfly's most common host plants are *Asclepias asperula*, *Asclepias incarnata*, *Asclepias oenotheroides*, *Asclepias syriaca*, and *Asclepias viridis* [8–11]. These plants are ecologically important, because in addition to supporting monarch butterflies, they harbor a diverse community of other arthropods including herbivores, nectarivores, transients, and predators [12,13]. At least 16 species of herbivorous arthropods specialize on milkweed plants [14]. *Asclepias tuberosa* flowers in Arizona were shown to harbor 80 different species of arthropods [15], and *Asclepias viridis* flowers in Oklahoma were visited by more than 23 families of arthropods [16]. In Texas, *Asclepias viridis* host plants were visited by 77 arthropod taxa in a spring study [13] and by 39 taxa in a fall study [12].

A critical feature affecting the milkweed host-plant community is variation in the cardenolide content of the arthropods that occupy these plants. As an anti-herbivory defense, milkweed plants produce toxic cardenolides [17,18]. Cardenolides are toxic because they interfere with the activity of Na$^+$/K$^+$-ATPase enzymes that regulate the transport of sodium and potassium across cell membranes [19,20]. However, not all organisms are equally affected by cardenolide exposure. These varying impacts were reviewed in detail by Agrawal et al., 2012 [19]. Many herbivorous arthropods are sensitive to cardenolides and avoid or do poorly on plants that contain them. Likewise, many predatory arthropods are also susceptible to cardenolide toxicity and avoid preying on arthropods that sequester cardenolides. On the other hand, some arthropods seem to tolerate cardenolides to varying extents. Several mechanisms can result in tolerance to cardenolides, including point mutations in the Na$^+$/K$^+$-ATPase enzyme that reduce the affinity of Na$^+$/K$^+$-ATPase to cardenolides [20,21], physiological and morphological attributes that prevent cardenolides from passing through the gut wall, and cardenolide metabolism [19]. Because of this, some species of arthropods seem to be able to avoid or eliminate cardenolides and, consequently, neither are affected by cardenolides nor sequester them [22]. Likewise, some arthropod predators are also able to tolerate cardenolide ingestion [19]. On the other hand, monarch butterfly larvae and several other arthropods that feed on milkweed sequester cardenolides and employ these sequestered cardenolides in their own defense against predators [17–19,23,24]. In short, the high abundance and diversity of arthropods on *Asclepias* plants and the varying abilities of these arthropods to sequester or tolerate cardenolides underlie a complex system of species interactions that are inherent to the milkweed host-plant community.

There are only a few studies on the survival and ecology of monarch butterflies during the spring generation, when the population is at its lowest. Estimates of spring-generation survival vary considerably [13,25–27]. Despite this variation, there is a consensus that most mortalities among spring eggs and larvae are due to arthropod predators, and much emphasis has been placed on mortality caused by red imported fire ants (RIFA) (*Solenopsis invicta*) [25,27–32]. In particular, two studies in Texas suggested that RIFA had devastating impacts on spring-generation monarch survival [25,27]. One of these studies recorded 100% mortality of monarch eggs and larvae and attributed this to RIFA [25]. However, information on the effects of RIFA on spring-generation monarchs are somewhat anecdotal [26] and based on small samples [25]. For example, the study that observed 100% mortality inferred that this was due to RIFA based on the abundance of RIFA on the study site and a single observation of a young instar being consumed by a RIFA [25]. Another study [26]

found high monarch mortality on some study sites and attributed this to observations of "ants" on the host plants. The third study experimentally manipulated RIFA density and documented lower survivorship where RIFA were excluded from host plants [27]. This study is intriguing, but it did not quantify how the manipulations (exclusion barriers) may have impacted other predators or the host-plant arthropod community. Our own observations indicate that RIFA will consume monarch larvae. However, as mentioned above, monarch eggs and larvae contain cardenolides, and the extent to which cardenolides affect the inclusion of monarch eggs and larvae in the diets of RIFA is unknown. In short, good quantitative data on the impacts of RIFA on monarch eggs and larvae are lacking. Most importantly, there are no quantitative data on how effective typical measures used to control RIFA might be in promoting spring-generation monarch survival.

RIFA are a globally invasive species native to South America [33–35]. They are believed to have first arrived in the USA in 1938 at Mobile, Alabama, through lumber shipments [36]. They first appeared in southeast Texas in 1953 and spread into north Texas by the late 1960s and early 1970s [37]. RIFA now occupy all of the core area of reproduction for spring-generation monarchs in Texas and much of the core area for spring-generation reproduction of monarchs in Oklahoma [35,38]. RIFA are well known for their negative impacts on a wide variety of taxa [39,40], including vertebrates [41,42] and invertebrates [39,43]. Because of these negative effects, the presence of RIFA can affect arthropod community structure and composition [43–47]. However, these effects seem to vary considerably and, in some cases, some types of arthropods benefit from the presence of RIFA [44,48]. In other cases, there appears to be no correlation between RIFA abundance and arthropod community diversity [49].

The variable effects of RIFA on arthropod populations are due to the foraging ecology of RIFA. RIFA are predominantly ground foragers [50], and non-volant ground-dwelling invertebrates are more likely to be negatively impacted by RIFA [50,51]. RIFA are also more likely to impact gregarious arthropods and utilize pheromone trails to exploit aggregating arthropods [52]. In addition, RIFA seek proteins and carbohydrates, and their foraging targets vary according to the current conditions of the colony [53], which in turn are affected by season, habitat, temperature, and colony size [52,54–56]. Thus, when the colony is deficient in proteins, RIFA seek high-protein food sources; when the colony is deficient in carbohydrates, RIFA seek food sources that are high in carbohydrates. Diet shifts related to carbohydrate availability and exploitation have been cited as particularly important determinants of the success of RIFA in their non-native distribution [57] and in their impact on arthropod communities [58] and prey selectivity [52]. The complexity of RIFA foraging behavior, how that foraging might be impacted by cardenolides, and how these factors are also affected by arthropod community structure mean that there might not be a simple correlation between the abundance of RIFA on a site and the impact of RIFA on monarch egg and larval survival. To evaluate the effect that RIFA have on monarch survival, a comprehensive experimental approach is necessary.

The overall purpose of this study was to conduct a detailed analysis of the role that RIFA have in the mortality of spring-generation monarch butterfly eggs to third instar. We start with a correlative approach and examine whether there is any evidence of a direct relationship between RIFA abundance on and around the host plants and monarch survival. Second, we use an experimental approach to manipulate the abundance of RIFA on or around the host plants to determine whether these manipulations impact monarch survival. Importantly, one of these manipulations, the suppression of RIFA populations around the host plants, specifically tests whether traditional methods used to control RIFA (broadcast baits) can be used as management techniques to improve monarch butterfly egg and larval survival. In another manipulation, we increased the abundance of RIFA on the host plants by placing mealworms on the host plants. In this way, we were able to evaluate the extent to which elevated RIFA abundance on host plants impacts monarch butterfly egg and larval survival. Lastly, in a previous study [13], it was found that monarch egg and larval survival is affected by other arthropods on the host plant. Here, we use the arthropod data from that

previous study [13] to determine how other host-plant arthropods affect the presence and abundance of RIFA on unmanipulated host plants, how these arthropods affect predator pressure on the host plant, and what the interrelationships are between predator pressure on the host plant, RIFA abundance on the host plant, and monarch survival.

## 2. Materials and Methods

The data for this study were collected at the Cooper Wildlife Management Area (829 CR 4795, Sulphur Springs, Hopkins Co., TX 75482, USA) and adjacent portions of Cooper Lake State Park (1690 FM 3505, Sulphur Springs, Hopkins Co., TX 75482, USA) (33°18′51.09″ N, 95°36′16.70″ W), and were collected from 21 March 2017 through 17 May 2017 and from 26 March 2018 through 11 May 2018. The only species of milkweed present on the study site was *Asclepias viridis*, and its density, measured in 2017, was 6540 plants per ha.

### 2.1. Monarch Survival and Host-Plant Arthropods

The methods used to monitor monarch survival and arthropod presence on host plants were described in detail in a previous paper [13] and generally follow protocols used by other authors [28,30,59,60]. Briefly, host plants were located by either searching for eggs on the plants or by watching female monarchs lay eggs. The location of the egg on the host plant was marked with a non-toxic marker, and the host plant itself was marked with a numbered flag. The eggs were visited daily between 10:00 h and 17:00 h. During these visits, the plants were approached carefully, noting any volant or mobile arthropods on the plant during approach. The status of the egg and all other arthropods was then determined. Once the eggs hatched, the larvae were monitored daily until they reached the third instar. We stopped at the third instar because other studies and our own observations showed that many larvae begin to leave their host plant once they reach the third instar [61–64]. Consequently, after the third instar, we could not discriminate between mortalities and emigration from the host plant. When an egg or larva was missing from the host plant, we revisited the host plant on the following four days to ensure that the egg or larva was not simply overlooked or temporarily off the host plant. Following the protocols of other studies [30,59,65], if the egg or larva was determined to be missing from the host plant, it was considered to have died.

We also measured the physical attributes of the host plants on the day after the egg was found and on the day that the larvae either reached the third instar or was determined to have died. Measurements were taken of the number of ramets, the length of each ramet from ground surface to the highest leaf or flower of each ramet, and the total number of mature leaves on each ramet. These measures were averaged between the first and last days of measurement to calculate the number of ramets, the combined length of all ramets, and the combined number of leaves for all ramets on the host plant.

### 2.2. RIFA Density and Abundance among Controls

We measured several aspects of RIFA abundance and density. For each host plant, in addition to daily monitoring of RIFA on the host plant, we measured the distance from the host plant to the nearest RIFA mound and the number of RIFA mounds within 4 m of the host plant. We also measured the above-ground volume of each RIFA mound by calculating $\frac{1}{2}$ of the volume of an ellipsoid based on the length, width, and height of the mound. These measures were used to calculate the total volume of RIFA mounds within 4 m of the host plant. All of these measures were made on the day after an egg was found and on the last day the egg or larva was monitored. The average of these two sets of measures was used for statistical analyses.

### 2.3. Experimental Manipulation of RIFA Abundance

We divided our 23 ha study area into six sub-plots designated for three treatment groups: two plots (5.3 ha and 4.4 ha) contained only control plants, two plots (2.1 ha and

5.3 ha) contained only RIFA-suppressed plants, and two plots (2.2 ha and 3.7 ha) contained only RIFA-enhanced plants. Each treatment plot was separated from other treatment plots by roadways, easements, and wooded areas so that there would be no spill-over effects among treatments.

The RIFA-suppressed treatment followed methods used in other studies [66–68]. We used Extinguish Plus Fire Ant Bait® (Central Garden & Pet Company, 1340 Treat Blvd, Suite 600, Walnut Creek, CA 94597, USA), composed of Hydramethylnon 0.365% and S-Methoprene 0.250% in a cornmeal carrier. This is a broadcast bait that is reported to have minimal impacts on non-target invertebrates [66,67]. Following the manufacturer's instructions, the bait was broadcast at a rate of 2.5 lbs per acre (2.8 kg/ha). RIFA baiting was never conducted during the field seasons. Rather, broadcast baiting, using hand spreaders, was conducted three times prior to the onset of the 2017 spring field season on 24 and 25 October 2016, 7 and 8 March 2017, and 20 March 2017. Baiting of RIFA was then repeated after the 2017 field season and prior to the 2018 field season on 27 June 2017, 12 October 2017, 3 March 2018, and 20 March 2018. During these times, individual mounds were also treated as they were encountered.

RIFA are notoriously hard to control, and though broadcast baiting was not conducted during the field seasons, individual mounds did occasionally appear over the course of the study in the treated plots. These mounds were treated with Bayer Advanced Fire Ant Killer Dust® (Bayer Environmental Science NA, 5000 CentreGreen Way, Cary, NC 27513, USA), which contains 0.5% β-Cyfluthrin. β-Cyfluthrin is sensitive to sunlight, and exposed treatments have a half-life of 48 to 72 h [69]. This treatment typically killed the ants within 24 h, and extreme care was taken to avoid exposure of the powder to surfaces other than the RIFA mound.

We also experimentally increased numbers of RIFA on host plants. For this purpose, we used Elmer's Wood Glue® (Elmer's Products, Inc., 460 Polaris Parkway Suite 5, Westerville, OH 43082, USA) to glue one freeze-dried mealworm on each of the four lowest leaves of each of the occupied host plants in the enhanced treatment plots. This practice was very effective, and the mealworms were quickly consumed by RIFA or, during rainy weather, were washed off the leaves. Consequently, the mealworms had to be replaced daily to keep the RIFA coming back onto the host plant.

*2.4. Statistical Analyses*

Statistical analyses were conducted using SAS® Studio 3.81 software (SAS Institute Inc., 100 Sas Campus Dr, Cary, NC 27513, USA). We placed emphasis on effect sizes but used *p*-values to evaluate the level of confidence associated with those effect sizes [70]. Data were tested for normality using the Shapiro–Wilk statistic and corroborated using the Kolmogorov D Statistic. Parametric tests were used where data were identified as normally distributed. Otherwise, we used non-parametric tests.

For the simple correlative analysis of RIFA parameters with monarch survival, a stepwise logistic regression procedure was used to determine whether monarch survival could be predicted from the distance to the nearest RIFA mound, the number of RIFA mounds within 4 m, and the total volume of RIFA mounds within 4 m. Also included in this analysis were the start date of monitoring, the number of ramets on the host plant, the total length of all ramets on the host plant, and the total number of leaves on all ramets on the host plant. The stepwise procedure produced a subset of models based on entry and removal probabilities. The best model was then identified from this subset using corrected Akaike Information Criterion scores (AICc) [71]. In presenting logistic regressions, we tabulate the entire subset of models and also present dAICc as the difference of each model from best model (lowest AICc score) and $w_i$ as the model weight for each model. The level of confidence (*p*-value) for each model is presented based on likelihood ratio Chi-square tests.

For the experimental data, we first evaluated how effective our treatments were in manipulating the presence of RIFA on and around the host plants. The data were not

normally distributed, so we used Kruskal–Wallis tests to compare the distance to nearest RIFA mound, number of RIFA mounds within 4 m, the volume of RIFA mounds within 4 m, and number of RIFA on plants for each treatment. We then use simple Chi-square contingency analyses to compare the survival of monarch eggs and larvae to the third instar among treatments. To further examine the potential differences between survival on control host plants and survival on RIFA-suppressed host plants, we conducted separate tests using 2 × 2 contingency tables and used the *p*-values from these tests to assess the level of confidence in the observed differences. In addition, we generated odds ratios based on binomial proportions to provide a measure of risk relative to controls. Odds ratios were calculated using the FREQ procedure in the SAS® software (SAS® Studio 3.81 software, SAS Institute Inc., 100 Sas Campus Dr, Cary, NC 27513, USA), which also provides asymptotic Wald confidence limits for the odds ratio estimate. These results were used to corroborate the results of the contingency table analyses.

For the host-plant arthropod data, the probability of an arthropod occurring on a given host plant was a function of the amount of time that the plant was monitored. Since host plants upon which mortalities occurred were monitored for fewer days than host plants upon which the eggs survived to the third instar, we had to correct for this bias. For this reason, for the arthropod analyses, plants monitored for less than 10 days were eliminated from the analyses to generate distributions with the same mean and variance for number of days monitored (see [13]). Since arthropod numbers might also be a function of host-plant size, we included total length of ramets, total number of leaves, and the number of ramets on the host plant as possible variables in the model selection process. In analyzing the arthropod data, we also had to deal with sparse and overdispersed distributions. While logistic regression models are relatively robust against deviations from normality [72], sparse data bias can lead to over-estimated effect sizes [73]. Consequently, as described in an earlier paper [13], we had to combine some arthropods into taxonomic and ecological groupings. Details on these groupings can be found in Stevenson et al. 2021 [13] and are duplicated here in Appendix A (Table A1) for reference purposes.

As described for the logistic regression analysis of RIFA parameters, a stepwise selection procedure was used to build predictive models of RIFA presence on control host plants. This analysis was based on 15 arthropod groups (see Appendix A, Table A1, for details on included taxa): aphids (Aphidoidea), leafhoppers (Hemiptera, Cicadomorpha), large milkweed bugs (*Oncopeltus fasciatus*), other milkweed herbivores (Arthropoda), weevils (Coleoptera, Curculionidae), leaf beetles (Coleoptera, Chysomelidae), dermestid beetles (Coleoptera, Dermestidae), other beetles (Coleoptera), mites (Arachnida, Acari), flies (Diptera), all other non-predatory arthropods, Little Black Ants (*Monomorium minimum*), other ants (Hymenoptera, Formicidae), jumping spiders (Araneae, Salticidae), and all predators other than ants and jumping spiders. To account for the effects of date and plant size, the variables available for model building also included the date monitoring began, the total number of ramets on host plant, the total length of ramets on host plant, and the total number of mature leaves on host plant. The results of the stepwise logistic regression procedure are presented as described earlier for the analysis of RIFA parameters.

The logistic regression analysis of arthropod groups identified taxa and groups of taxa that predicted the presence of RIFA on host plants. However, that analysis did not fully evaluate how the abundance of arthropods on the host plants affected the number of RIFA on the host plant, the predator pressure on the host plant, or the interactions among arthropod abundance, RIFA abundance, predator pressure, and monarch survival. To obtain a better idea of these interactions, we divided the total number of arthropods other than RIFA on the host plants into 8 abundance classes, each with about 30 observations. For each abundance class, we then calculated the mean number of RIFA on the host plants, the percentage of arthropods (including RIFA) that were predators (a crude measure of predator pressure), and the percent of monarch eggs that survived to the third instar. This approach allowed us to investigate how RIFA responded to other arthropods on the host

plant, how this response affected predator pressure, and how these parameters affected monarch survival.

## 3. Results

Data were collected on 886 eggs. In 2017, 191 eggs on 130 host plants were on control plots, 107 eggs on 65 plants were on suppressed plots, and 85 eggs on 65 plants were on enhanced plots. In 2018, 257 eggs on 150 plants were on control plots, 126 eggs on 74 plants were on suppressed plots, and 120 eggs on 77 plants were on enhanced plots. Based on the number of RIFA mounds within 4 m of the control plants, the overall density of RIFA on our study site was $630.51 \pm 29.03$ (n = 191) mounds per ha in 2017. RIFA density was lower in 2018 when the estimated density was $527.93 \pm 16.68$ (n = 257) mounds per ha (t-test, t = 3.23, df = 446, $p$ = 0.0013).

### 3.1. Analyses of RIFA Measures and Monarch Survival among Controls

The stepwise logistic regression procedure failed to find any RIFA variables that predicted monarch survival (Table 1). The only important variable was a positive effect of the number of ramets on monarch survival. Interestingly, the model that included the number of RIFA mounds and the distance to the nearest RIFA mound indicated that survival was positively associated with the number of mounds and negatively associated with the distance to the nearest mound (i.e., more mounds = higher survival, closer mounds = higher survival). However, these effect sizes were extremely small, and these parameters were associated with a great deal of uncertainty (maximum likelihood estimates, $p$ = 0.5035 and $p$ = 0.4355, respectively).

**Table 1.** Stepwise logistic regression of RIFA variables used to predict the survival of monarch eggs and larvae on control plants. A subset of models was generated using criteria for variable entry set at a $p$-value of 0.50 and for variable removal set at a $p$-value of 0.55.

| Model | AICc | dAICc | $w_i$ | Likelihood Ratio $X^2$ | Model Probability |
|---|---|---|---|---|---|
| Intercept, Number of Ramets | 350.592 | 0.000 | 0.487 | 6.2681 | 0.0123 |
| Intercept, Number of Ramets, Number of RIFA Mounds | 351.585 | 0.993 | 0.297 | 7.3027 | 0.0260 |
| Intercept, Number of Ramets, Number of RIFA Mounds, Distance to Nearest RIFA Mound | 352.843 | 2.251 | 0.158 | 8.0805 | 0.0444 |
| Intercept Only | 354.842 | 4.250 | 0.058 | - | - |
| **Summary of best-fit model. Concordance of this model was 49.9 percent.** | | | | | |
| **Parameter** | **DF** | **Estimate** | **Standard Error** | **Wald Chi-Square** | ***p*** |
| Intercept | 1 | −2.3582 | 0.2461 | 91.8391 | <0.0001 |
| Number of Ramets | 1 | 0.1980 | 0.0760 | 6.7878 | 0.0092 |

### 3.2. Effect of Experimental Manipulation of RIFA on Measures of RIFA Abundance

As expected, drawing RIFA onto the host plants (enhanced treatment) did not affect the distance to the nearest RIFA mound or the number of RIFA mounds within 4 m of the host plants (Table 2). However, there were over 21 times as many RIFA on plants in the enhanced treatment as were found on plants in the control treatment (Table 2). Only 1 of the 114 eggs evaluated in the enhanced treatment did not have RIFA on its host plant. The suppressed treatment had marked effects on all measures of RIFA abundance. Compared to the control plants, host plants in the suppressed treatment were over 9 times farther from the nearest RIFA mound, had 13 times fewer RIFA mounds within 4 m, had almost 5 times lower total volume of RIFA mounds within 4 m, and had 70 times fewer RIFA observed (Table 2). Only 5 of the 160 eggs evaluated in the suppressed treatment were observed to have RIFA on their host plants.

**Table 2.** Effects of treatments on measures of RIFA abundance. Note that the number of RIFA on host plants was corrected for observational bias by eliminating host plants observed for less than 10 days (see methods).

| Parameter | Suppressed Treatment Mean ± SE (n) | Control Mean ± SE (n) | Enhanced Treatment Mean ± SE (n) | Kruskal–Wallis Test $X^2$ (df = 2) | $p$ |
|---|---|---|---|---|---|
| Distance to Nearest Mound (cm) | 1920.72 ± 102.54 (233) | 208.30 ± 9.79 (448) | 222.35 ± 11.51 (205) | 434.2832 | <0.0001 |
| Number of Mounds ≤ 4 m from Host Plant | 0.21 ± 0.03 (233) | 2.87 ± 0.08 (448) | 2.41 ± 0.10 (205) | 448.9712 | <0.0001 |
| Total Volume of Mounds ≤ 4 m from Host Plant (cm³) | 4817.11 ± 1674.09 (233) | 23,485.36 ± 1562.52 (448) | 26,182.02 ± 2994.29 (205) | 390.6882 | <0.0001 |
| Number of RIFA on Host Plant | 0.04 ± 0.02 (160) | 2.83 ± 0.67 (224) | 60.97 ± 3.63 (114) | 342.8847 | <0.0001 |

*3.3. Effect of Experimental Manipulation of RIFA on Monarch Survival*

The effects of the RIFA treatments on the survival of monarch eggs to the third instar varied considerably when both years were combined and when each year was considered separately (Figure 1). This was due to the fact that drawing RIFA onto host plants (enhanced treatment) strongly decreased monarch survival. For both years combined, when the enhanced treatment was removed from the analysis, the difference between control and suppressed treatments was less pronounced. In particular, in 2017, there was little difference between the control treatment and the suppressed treatment. However, in 2018, there appeared to be lower survival within the control treatment than was observed among eggs in the suppressed treatment (Figure 1).

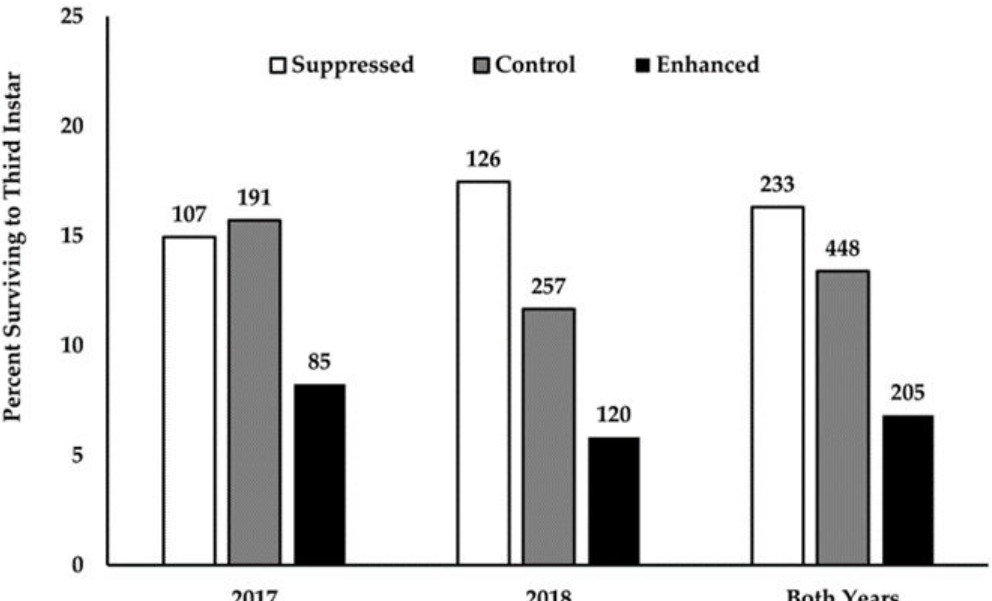

**Figure 1.** Effect of RIFA treatments on survival of monarch eggs and larvae to the third instar in north Texas. Numbers over bars represent the number of eggs in each treatment. There were differences in survival among treatments when both years were combined (Chi-square, 2 × 3 Contingency Table, $X^2$ = 9.3382, df = 2, p = 0.0094). For both years combined, when the enhanced treatment was removed from the analysis, the differences between control and suppressed treatments were slight (Chi-square, 2 × 2 Contingency Table, $X^2$ = 1.0581, df = 1, p = 0.3037).

We examined the differences between controls and suppressed treatments in greater detail using odds ratios (Table 3).

**Table 3.** Comparisons of survival to third instar among eggs on control plants and eggs on plants where RIFA were suppressed. Numbers in parentheses are the sample sizes for each treatment. Where the odds ratio is greater than one, then survival is lower for controls.

| | Survival (%) | | Chi-Square (1 df) | *p* | Odds Ratio | 95% Confidence Interval |
| | Control | Suppressed | | | | |
|---|---|---|---|---|---|---|
| 2017 | 15.71 (191) | 14.95 (107) | 0.0298 | 0.8629 | 0.9436 | 0.4882–1.8237 |
| 2018 | 11.67 (257) | 17.46 (126) | 2.4133 | 0.1203 | 1.6006 | 0.9763–2.9081 |
| Both Years | 13.39 (448) | 16.31 (233) | 1.0581 | 0.3075 | 1.2602 | 0.8106–1.9592 |

In 2017, there was little difference between the control and the suppressed treatments relative to survival to the third instar, where survival among controls was 15.71% and survival in the suppressed treatment was 14.95% (Table 3). In that year, based on the odds ratio, control eggs were 0.94 times as likely to die as eggs in the supressed treatment. In 2018, survival to third instar among eggs in the suppressed treatment was 17.46%, whereas among eggs on control plants, survival was 11.67%. Based on the odds ratio, eggs on host plants in the control treatment were 1.6 times more likely to die than eggs on host plants in the suppressed treatment. However, the confidence interval for the odds ratio is large, and the *p*-value associated with this difference is fairly high, indicating a low level of confidence in this difference (Table 3).

### 3.4. What Attracts RIFA onto Host Plants and How Does That Impact Monarch Survival?

Among the 448 eggs on control plants, less than 30% (128) were observed to have RIFA on their host plant. However, our enhanced treatment demonstrated that RIFA could be drawn onto host plants and that doing so negatively impacted monarch survival. We wanted to know what conditions predicted the presence of RIFA on host plants and how the abundance of RIFA might affect monarch survival to the third instar.

Stepwise logistic regression, based on the arthropod groups associated with 224 eggs on control host plants, generated six predictive models (Table 4). The best model included six groups of arthropods, all of which had positive effects in predicting the presence of RIFA on the host plant. Among non-predatory taxa, aphids, weevils, and leaf beetles predicted the presence of RIFA, though the strongest effects were associated with weevils and leaf beetles. The presence of RIFA was also predicted by the presence of predatory taxa, mainly jumping spiders and the nebulous group of "all predators other than ants and jumping spiders" (Table 4).

**Table 4.** Stepwise logistic regression analysis used to predict RIFA presence on control host plants based on arthropod groups found on those host plants. A subset of models was selected using criteria for variable entry set at a *p*-value of 0.20 and for variable removal set at a *p*-value of 0.25.

| Model | AICc | dAICc | $w_i$ | Likelihood Ratio $X^2$ | Model Probability |
|---|---|---|---|---|---|
| Jumping Spiders, Weevils, Predators other than Ants and Jumping Spiders, Leaf Beetles, Ants other than RIFA or Little Black Ants, Aphids | 205.2060 | 0.0000 | 0.6249 | 85.4613 | <0.0001 |
| Jumping Spiders, Weevils, Predators other than Ants and Jumping Spiders, Leaf Beetles, Ants other than RIFA or Little Black Ants | 206.3443 | 1.1383 | 0.3537 | 82.3591 | <0.0001 |
| Jumping Spiders, Weevils, Predators other than Ants and Jumping Spiders, Leaf Beetles | 211.9911 | 6.7851 | 0.0210 | 74.7674 | <0.0001 |
| Jumping Spiders, Weevils, Predators other than Ants and Jumping Spiders | 219.9316 | 14.7256 | 0.0004 | 64.8995 | <0.0001 |
| Jumping Spiders, Weevils | 230.3222 | 25.1162 | 0.0000 | 52.6019 | <0.0001 |
| Jumping Spiders | 251.1491 | 45.9431 | 0.0000 | 29.8874 | <0.0001 |
| Intercept Only | 279.1675 | 73.9615 | 0.0000 | - | - |

Table 4. *Cont.*

| Model | AICc | dAICc | $w_i$ | Likelihood Ratio $X^2$ | Model Probability |
|---|---|---|---|---|---|
| **Summary of the best-fit logistic regression model. Concordance of this model was 83.6%.** | | | | | |
| Parameter | DF | Estimate | Standard Error | Wald Chi-Square | *p*-value |
| Intercept | 1 | −2.3358 | 0.2879 | 65.8196 | <0.0001 |
| Aphids | 1 | 0.0016 | 0.0011 | 2.0962 | 0.1477 |
| Ants other than RIFA or Little Black Ants | 1 | 0.1870 | 0.1281 | 2.1317 | 0.1443 |
| Weevils | 1 | 0.1276 | 0.0402 | 10.0552 | 0.0015 |
| Predators other than Ants and Jumping Spiders | 1 | 0.3617 | 0.1598 | 5.1245 | 0.0236 |
| Jumping Spiders | 1 | 0.3089 | 0.1368 | 5.0988 | 0.0239 |
| Leaf Beetles | 1 | 0.3223 | 0.1124 | 8.2211 | 0.0041 |

Monarch survival increased with increasing numbers of arthropods (not including RIFA) on the host plant, but predator pressure remained relatively constant (Figure 2). RIFA did not increase in abundance until there were large numbers of other arthropods on the host plant, and the presence of large numbers of RIFA did not affect predator pressure. Because of this, there was no correlation between monarch survival and RIFA abundance (Pearson's Correlation Coefficient, r = 0.54765, *p* = 0.1600) or between monarch survival and predator pressure (Figure 2) (Pearson's Correlation Coefficient, r = 0.35042, *p* = 0.3948).

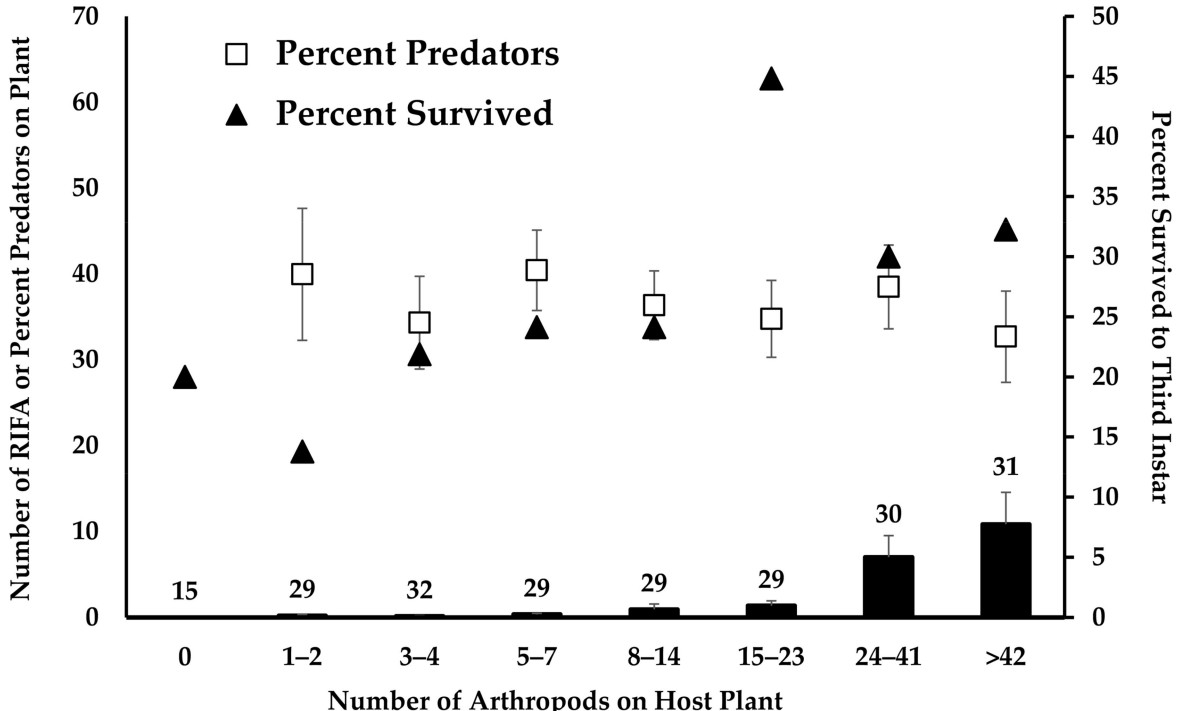

**Figure 2.** Effect of other host-plant arthropods on monarch survival, predator pressure, and RIFA abundance. The percent survival of monarch eggs to third instar (triangles) increased as host-plant arthropod abundance increased (Mantel–Haenszel Chi-Square test, $X^2$ = 4.7193, df = 1, *p* = 0.0299). The percent of predatory arthropods on the host plant (open squares) did not vary relative to the abundance of other arthropods (not including RIFA) on the host plant (Kruskal–Wallis test, $X^2$ = 2.5094, df = 6, *p* = 0.8674). RIFA abundance (bars) remained low until the number of other arthropods on the host plant exceeded 24 (Kruskal–Wallis test, $X^2$ = 49.4566, df = 6, *p* < 0.0001). Numbers over bars are sample sizes for each host-plant arthropod frequency class.

## 4. Discussion

As reported in a previous study [13], monarch survival from egg to third instar on our study site was high despite the fact that the RIFA mound density on the study site (631–528 mounds/ha) was considerably greater than the average densities of RIFA reported for the southeastern USA. (155–470 mounds/ha) [74,75]. Among control plants, we could find no clear relationship between monarch survival to the third instar and the distance of the host plant to the nearest RIFA mound, the number of RIFA mounds within 4 m of the host plant, or the total volume of RIFA mounds within 4 m of the host plant.

The use of a targeted bait resulted in significant suppression of RIFA abundance as evidenced by the large reduction in mound density, volume, and distance from the host plant. Most importantly, this treatment seemed to largely eliminate RIFA from the host plants. Despite this, across both years of the study, treating plots to control for RIFA only increased survival by just under 3%. In 2017, survival among control host plants was, if anything, slightly higher than it was on host plants in the suppressed treatment. However, in 2018, monarch eggs in the suppressed treatment were 1.6 times more likely to survive to the third instar than eggs in the control treatment. Though this difference was associated with a low level of confidence ($p = 0.1203$), it does suggest that in some years, RIFA could have a greater impact on monarch survival than in other years. The reason for this is not evident from the data collected in the current study, especially since RIFA mound density was lower in 2018 than it was in 2019. It may be important that 2017 was warmer and wetter than normal, whereas 2018 was much cooler and drier than normal (NOAA National Centers for Environmental Information, Sulphur Springs, Texas, weather monitoring station). These deviations had a noticeable impact on the phenology of milkweed development and arthropod emergence and may have impacted the availability of RIFA prey or the dietary demands of RIFA for proteins versus carbohydrates [54,76–79]. Protein and carbohydrate intake by RIFA colonies are affected by the nutritional status of the colonies [53] and habitat [52]. Importantly, weather conditions can also affect the diets of RIFA colonies [54–56]. In particular, wet weather such as that observed in 2017 results in RIFA workers foraging for more carbohydrates relative to proteins [55].

On control plants, monarch survival increased with increasing numbers of other arthropods (not including RIFA) on the host plant, consistent with previous findings on this study site [13]. However, predator pressure, as measured by the percentage of predatory arthropods (including RIFA) on the host plant, was essentially constant relative to the number of other arthropods on the host plant. Furthermore, RIFA abundance did not track arthropod numbers on host plants until there were large numbers of other arthropods on the host plant. The combination of the low incidence of RIFA on control plants with few arthropods and a lack of an effect of RIFA on predator pressure likely accounts for the observation that monarch mortality was unrelated to the number of RIFA on control host plants.

We believe that the results identified above are a reflection of the foraging ecology of RIFA and help to explain why suppressing RIFA was, at best, minimally effective. Several factors are important. First, RIFA are predominantly ground foragers [50] and do not ascend milkweed plants in large numbers unless there are also large numbers of arthropods on the plant. Second, because monarch eggs and larvae up to the third instar are small and highly dispersed, they are unlikely to be specific search targets for RIFA scouts [50,51]. RIFA in Oklahoma fed most frequently on arthropods that were either social species or formed aggregations in suitable microhabitats [52]. Consequently, the predation of monarch eggs and early instar larvae by RIFA is most likely opportunistic. Third, as mentioned earlier, RIFA vary their diet relative to carbohydrate and protein intake depending on the demands of the colony. As a result, when RIFA do ascend a host plant, they may prefer some types of prey over others [76]. Our logistic regression analysis indicated that RIFA were most likely to ascend host plants that contained aphids, weevils, and leaf beetles. Aphids and weevils are phloem feeders and, as reported elsewhere [79], may be favored by RIFA for their high carbohydrate content. Most of the leaf beetles observed on the host plants were on flowers

and appeared to be seeking nectar or pollen [80] and would also be high in carbohydrates. In addition, host plants with numerous RIFA harbored other predators, likely also drawn to the prey taxa present, and these predators also serve as prey items for RIFA [67]. Lastly, it may be important that, unlike monarch eggs or larvae, the arthropods identified by the logistic regression as predicting the presence of RIFA on the host plant are unlikely to contain high levels of toxic cardenolides [22], possibly making them more desirable as prey items. It is known that at least some arthropod predators avoid consuming monarch eggs and larvae owing to the presence of cardenolides [81]. Studies with other species of ants and their associations with aphids demonstrate that when given a choice, the ants are more likely to associate with aphids that have lower concentrations of cardenolides [82,83]. It would be interesting to test the selective behavior of RIFA when given a choice between monarch eggs or caterpillars and other prey types that do not contain cardenolides.

The factors described above may explain why survival was high on control plants and why suppressing RIFA had minimal impacts on monarch survival. On the other hand, gluing mealworms onto the bottom leaves of host plants markedly increased the number of RIFA on the host plants and also markedly increased monarch mortality. This stands in contrast to the control plants, where large numbers of RIFA were not associated with increased monarch mortality. We believe that the enhanced treatment altered the general relationship between the number of RIFA on the host plants, the total number of arthropods, and predator pressure. This is because the mealworms were continually replaced and consequently represented a constant and predictable food resource. Such conditions favor the establishment of pheromone trails for forager recruitment [50,84] and local area searching by scouts in the vicinity of the mealworms [51]. The end result was that host plants in the enhanced treatment had, on average, over 21 times as many RIFA on them. Likewise, the predator pressure on enhanced host plants was 2.4 times higher than it was on control plants. We think that these dramatically altered densities of RIFA and high levels of predator pressure account for the high mortality observed on the RIFA-enhanced host plants.

Two earlier studies, both conducted in Texas, have specifically identified RIFA as having catastrophic effects on monarch survival. Calvert 1996 [25] reported 100% mortality of monarch eggs and larvae and attributed this to the presence of RIFA. In a follow-up study, Calvert 2004 [27] used enclosures to isolate host plants from RIFA in order to measure the effects of RIFA on monarch survival. Based on the data presented in that paper, survival to the third instar varied from 3.1% to 11.9% inside the enclosures and from 0% to 0.5% outside the enclosures. A third study, by Lynch and Martin [26], conducted in Louisiana and north Texas, suggested that spider and ant predation resulted in an average of only 3% of eggs reaching the third instar [26]. Collectively, these three studies indicate lower survivorship than was found in the current study and allude to the role of RIFA as a source of this lower survivorship.

We have discussed how methodological issues might account for the differences in survival estimates among studies elsewhere [13]. However, the data presented in the current study point to another important reason why monarch survivorship might vary among studies and study sites. Our study site was a wildlife management area specifically managed for high plant diversity, which, in turn, leads to high arthropod abundance and diversity [85,86]. This high arthropod abundance and diversity seemed to buffer the effects of RIFA on monarch survival. The studies conducted by Calvert [25,27] and by Lynch and Martin 1993 [26] all occurred in pastures. Pastures in north Texas and Louisiana vary enormously according to management intensity, and they range anywhere between monocultures of Bahia grass (*Paspalum notatum*) or Bermuda grass (*Cynodon dactylon*) and much more diverse old-field habitats. This variation in biodiversity could explain why Lynch and Martin [26] found that survival from egg to third instar varied from 0% to over 30% among the six study sites that they used. Clearly, this is an area of spring-generation monarch conservation ecology that merits further study on multiple sites of varying biodiversity.

Effective control of RIFA is costly and extremely labor intensive. A long-term study based on the methods used in the current study would shed greater light on the efficacy of RIFA control for monarch survival. However, our data, despite some interannual variation, suggest that RIFA control may not be necessary. Rather, management for arthropod diversity, mediated through enhanced plant diversity, would be an effective and less costly long-term method to promote spring monarch survival. Such a management strategy, in turn, will help mitigate global declines in other arthropods [87–90]. Our current data are particularly relevant to larger nature preserves and management areas whose mission already is the promotion and maintenance of biodiversity. Small garden plots, small butterfly way-stations, or plots with low plant diversity are more likely to require RIFA control, because in these contexts, small plot sizes and lower arthropod diversity within and surrounding these sites will exacerbate the negative effects of RIFA on monarch survival.

**Author Contributions:** Conceptualization, J.G.K.; methodology, K.L.H., M.S., A.S., K.C. and J.G.K.; formal analysis, K.L.H. and J.G.K.; investigation, K.L.H., M.S., A.S., K.C. and J.G.K.; resources, J.G.K.; data curation, J.G.K.; writing—original draft preparation, K.L.H.; writing—review and editing, K.L.H., M.S., A.S., K.C. and J.G.K.; visualization, K.L.H. and J.G.K.; supervision, J.G.K.; project administration, J.G.K.; funding acquisition, J.G.K. All authors have read and agreed to the published version of the manuscript.

**Funding:** This research was funded by the Texas Comptroller of Public Accounts, Economic Growth and Endangered Species Management Division, Contract Numbers 5975LV and 6192CS. In-kind matching funds were provided by the College of Science and Engineering, Texas A&M University—Commerce.

**Data Availability Statement:** The data presented in this study are available on request from the corresponding author.

**Acknowledgments:** We would like to thank the numerous field assistants who aided in the col-lection of field data: Emily Casper, Nathan Connon, Hannah Dill, Nikki Dawson, and Beth Fortner. Thanks are also extended to Howard Crenshaw, TPWD Wildlife Division, for assistance working on the Cooper Wildlife Management Area and to Kody Waters for his assistance with working on the Cooper Lake State Park property. Mike Quinn, at The University of Texas at Austin, is thanked for assistance in arthropod identification.

**Conflicts of Interest:** The authors declare no conflict of interest. The funders had no role in the design of the study; in the collection, analyses, or interpretation of data; in the writing of the manuscript; or in the decision to publish the results.

## Appendix A

**Table A1.** Fifteen arthropod groups used in the logistic regression analysis to predict RIFA occurrence on control host plants associated with 224 eggs. Predatory taxa are highlighted in yellow. For full details on arthropods detected in the study and how these groups were defined, please consult Stevenson et al., 2021 [13].

| Common Name | Included Taxa | Abundance | Frequency | % Frequency |
|---|---|---|---|---|
| Aphids | Aphids (Aphididae) | 10,792 | 80 | 35.71 |
| Other Ants | Hymenoptera, Formicidae, Others | 907 | 37 | 16.52 |
| Little Black Ant | Hymenoptera, Formicidae, *Monomorium minimum* | 855 | 74 | 33.04 |
| Weevils | Coleoptera, Curculionidae | 471 | 67 | 29.91 |
| Mites | Arachnida, Acari, Mites | 268 | 62 | 27.68 |

**Table A1.** *Cont.*

| Common Name | Included Taxa | Abundance | Frequency | % Frequency |
|---|---|---|---|---|
| Other Non-Predatory Arthropods | Stick Insects (Phasmatodea), Crickets (Gryllidae), Click Beetles (Elateridae), Darkling Beetles (Tenebrionidae), Leaf-footed Bugs (Coreidae), Seed Bugs (Lygaeidae), Plant Bugs (Miridae), Unidentified Wasps (Apocrita), Millipedes (Diplopoda), Springtails (Collembola), Ticks (Acari), Shield Bugs (Pentatomoidea, non-predatory), Butterflies, Skippers, and Moths (Lepidoptera), Caddisflies (Trichoptera), Slugs and Snails (Mollusca), Mayflies (Ephemeroptera), Harvestmen (Opiliones), Bees (Hymenoptera, Apidae), Grasshoppers (Caelifera), Katydids (Tettigoniidae), Unidentified True Bugs (Hemiptera, Heteroptera) | 267 | 111 | 49.55 |
| Jumping Spiders | Araneae, Salticidae | 247 | 116 | 51.79 |
| Other Predatory Arthropods | Rove Beetles (Staphylinidae), Soldier Beetles (Cantharidae), Ground Beetles (Carabidae), Assassin Bugs (Reduviidae), Predatory Stink Bugs (Pentatomidae, Asopinae), Vespid Wasps (Vespidae), Scorpionflies (Mecoptera), Lacewings (Neuroptera), Hoverflies (Syrphidae), Ladybeetles (Coccinellidae), Wolf Spiders (Lycosidae), Grass Spiders (Agelenidae), Nursery Web Spiders (Pisuridae), Long-jawed Orb Weavers (Tetragnathidae), Lynx Spiders (Oxyopidae), Crab Spiders (Thomisidae), Unidentified Spiders | 227 | 123 | 54.91 |
| Other Leaf Beetles | Coleoptera: Flea Beetles (Chysomelidae, Alticini) and all other Leaf Beetles (Chysomelidae) | 171 | 65 | 29.02 |
| Dermestid Beetles | Coleoptera, Dermestidae | 139 | 28 | 12.50 |
| Flies | Midge Flies (Chironomidae), Fruit Flies (Drosophilidae), Mosquitoes (Culicidae), and Unknown Flies | 138 | 81 | 36.16 |
| Leafhoppers | Hemiptera, Cicadomorpha | 137 | 77 | 34.38 |
| Large Milkweed Bugs | Hemiptera, Lygaeidae, *Oncopeltus fasciatus* | 108 | 38 | 16.96 |
| Other Milkweed Herbivores | Small Milkweed Bugs (*Lygaeus kalmia*), Milkweed Longhorn Beetles (*Tetraopes texanus*), and Thrips (Thysanoptera) | 48 | 34 | 15.18 |
| All Other Beetles | Unidentified Beetles (Coleoptera) | 33 | 22 | 9.82 |

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
