# Peer review of "Experimental Suppression of Red Imported Fire Ants (Solenopsis invicta) Has Little Impact on the Survival of Eggs to Third Instar of Spring-Generation Monarch Butterflies (Danaus plexippus) Due to Buffering Effects of Host-Plant Arthropods"

_diversity, doi:10.3390/d15030331_

Round 1

Reviewer 1 Report

Hudman et al. experimentally investigated the complex interaction between imported red fire ants and the spring generation's offspring of the monarch butterfly. They found little evidence for a specific effects of fire ants on the survival of the monarch offspring, but in contrast show that high arthropod abundance on the host plants resulted in increased survival of the monarchs.

In general, the study provides a nice amount of data and shows a novel insight into the ecological relevance of fire ants as neozoon. The main outcome that high arthropod settlement decreases predation risk of monarch offspring by fire ants mertis publication. However, the manuscript requires sustantial improvement to be accepted for publication:

Main issues:

-title: based on the results of the study, the title should be reconsidered. The title right now is somewhat meaningless, exept for the fact that something was investigated for which both fire ants and monarchs are of concern: please be more specific.

-Abstract: the abstract lacks any information of the backround of the study. It should provide guidence into the topic and its relevance in the first 2-3 sentences.

-Introduction: the introduction is too short and does not provide necessary information of concern for the manuscript. To enable the reader to find the significance and relevance of the experiments, it needs more information on

a) monarchs and their biology. Why do they migrate, when and where? What's the relevance of the fields investigated in the manuscript for their life cycle?

b) the interaction between fire ants and monarchs: it is stated that fire ants have a negative effect on monarchs, but what is known on the relation between these two taxa? Do fire ants have dietary preferences? Do they take eggs or larvae, or both? Is this of sesonal relevance?

c) The actual aims of the study should be presented more concise to highlight what has been done.

Furthermore, the Material and Methods section includes several aspects of relevance for the introduction. Several passages describe the reasons for the experiments in the methods. The overall information on fire ant control, monitoring and monarch survival are of concern for the introduction already.

-Material and Methods: The Material & MEthods section is quite long and the information is presented in one large paragraph. Please, divide the text into sections and assign subheadings to them to guide the reader through the Methods.

Minor issues:

-keywords: please use keywords, that are unique from the title. The title is indexed anyway, therefore keywords should be chosen to supplement visibility of the article in addition to the title; also: "Solenopsis Invicta" should have a minor letter in the epithet.

-consistency: the fire ants are written differently in different places: "Red Imported Fire Ants" vs. "red imported fire ants", please be consistent

-Abstract: "fire ants (fire ants)" appears redundant

-Material and Methods: 

Please indicate facturers/brands for devices and reagents used.

How was parametricity assessed for the statistics?

-Results: 

-species names should be in italics

-3.0: Fire ant density was compared via an ANOVA. Why did the authors not use a t-test?

-Numbers and SD, e.g. 3.0 density of fire ants: mean and SD should have the same significant decimals. E.g. if the SD has two decimals, the mean should display the same amount of decimals.

-Conclusion: a conclusion would be helpful

Author Response

Our original intention was to be more succinct. However, your suggestions are good ones and we have added a considerable amount of additional detail on the ecology of the monarch butterfly, milkweed arthropod communities, and red imported fire ants to the introduction. We think this has made the manuscript much more comprehensive and has improved it considerably.

Reviewer 2 Report

Overall interesting, seemingly sound and well-presented.

Some points

Using data from a published study is probably ok, but teh corresponding phrase in the intro ("that … data")  ccome even clearer, plus should be followed by a citation. 

Abstract and intro

"Predators" - are probably always "predators other than fire ants". This is not quite clear in Abstract. Same with "arthropods" - which probably don't include fire ants as well.

Chemical defense of monarch larvae - should probably already be mentioned in Intro?

Another point - are there any behavioural studies/observations of (fire) ant predation on monarch larvae. Could be interesting to know whether ants successfully/easily attack healthy larvae.

Figure 1 is missing - I blame the journal for not checking.

chapter 3.4: useing = using

Author Response

We very much appreciate you time and consideration. We hope we have adequately addressed  your concerns.

Reviewer 3 Report

The captions of all the tables are very extensive. They should focus on what is shown and place the rest in what appears to be part of the material and methods, avoiding duplication of content.

The first part of the paragraph of point 3.1 seems to correspond more to the material and methods section than to the results.

Something similar happens in 3.2, 3.3 and 3.4, where much of the commented text can be specified in material and methods.

At the end of point 3.2 of results, it is indicated that "Only five of the 160 eggs evaluated in the suppressed treatment were observed to have fire ants on their host plants", however, in the first paragraph of results it is indicated that there are 233 eggs those evaluated for that treatment in both sampling years.

Figure 1 is not shown in the article and cannot be reviewed. It’s probably an editing error. Even so, note that relevant information is provided in the caption when it should appear in the results and not here.

In point 3.3, where it says "15.6%" it should say "15.71%".

The explanatory paragraph of the data shown in table 4 is difficult to understand, it should be redrafted.

Author Response

Thank you for the time and effort spent on this review. We found that your comments were constructive and have tried to address them as comprehensively as possible. 

Round 2

Reviewer 1 Report

The authors spent a decent amount of work to improve the understandability of the article and amended the crucial points of concern. It can be accepted for publication.